# Initial Study on the Impact of Probiotics on Postoperative Gastrointestinal Symptoms and Gut Microbiota after Sleeve Gastrectomy: A Placebo-Controlled Study

**DOI:** 10.3390/nu16203498

**Published:** 2024-10-15

**Authors:** Natalia Dowgiałło-Gornowicz, Dominika Mysiorska, Ewelina Sosnowska-Turek, Anna Botulińska, Paweł Lech

**Affiliations:** 1Department of General, Minimally Invasive and Elderly Surgery, Collegium Medicum, University of Warmia and Mazury, 10-045 Olsztyn, Poland; dominikamysiorska@gmail.com (D.M.); lechpawel@op.pl (P.L.); 2Probios Ltd., 10-683 Olsztyn, Poland; ewelina.sosnowska-turek@probios.com.pl; 3Department of Family Medicine and Infectious Disease, Collegium Medicum, University of Warmia and Mazury, Warszawska 30 St., 10-082 Olsztyn, Poland; annabotulinska@gmail.com

**Keywords:** bariatric surgery, probiotics, sleeve gastrectomy, constipation, nutrition

## Abstract

Background: Sleeve gastrectomy (SG) has become the predominant bariatric surgery, leading to significant weight loss and reductions in obesity-related complications. However, postoperative gastrointestinal symptoms such as constipation and bloating are common. This study aims to evaluate the impact of probiotic supplementation on postoperative gastrointestinal symptoms in patients undergoing SG. The secondary aim is to analyze laboratory and stool test results. Materials and methods: This prospective, placebo-controlled study included patients undergoing SG at a single center. Participants were adults without specific gastrointestinal diseases. They were randomly assigned to either the Probiotics or Controls group. Gastrointestinal symptoms and laboratory and stool tests were assessed before surgery and one month after. Results: Thirty-one patients participated, with 15 in the Probiotics group and 16 in the Controls group. Probiotic supplementation significantly increased the number of stools per week (*p* = 0.027) and reduced constipation incidence (*p* = 0.002). Patients in the Probiotics group reported easier defecation and greater bowel movement completeness (*p* = 0.015, *p* = 0.004). No significant differences in weight loss or laboratory tests were observed between the groups. Stool microbiota analysis showed a return to normal levels of *Enterococcus faecalis*, *Enterococcus faecium*, and *Clostridium perfringens* in the Probiotics group and an increase in the Controls group. Conclusions: Probiotic supplementation after SG significantly reduces constipation without adverse effects. These findings suggest that incorporating probiotics into postoperative care protocols can enhance patient comfort and recovery.

## 1. Introduction

According to the WHO, approximately 15% of adults worldwide suffer from obesity, and this number is still increasing [1]. Over the past decade, sleeve gastrectomy (SG) has become the dominant bariatric surgery procedure, and it has been shown to lead to significant weight loss and reduction in obesity-related complications [2,3,4]. Despite benefits, surgery is associated with postoperative gastrointestinal complications, which can cause symptoms such as constipation, abdominal pain, bloating, and malabsorption of micro- and macronutrients [5,6,7].

Moreover, obesity is associated with reduced gut microbiota (GM) diversity and a high rate of micronutrient deficiency. Bariatric surgery leads to significant changes in the composition and functions of the GM, which may impact micronutrient status. GM is not fully restored after bariatric surgery; therefore, probiotic supplementation represents a promising therapeutic approach in these patients [8,9].

Previously published papers on this issue demonstrate the positive effects of probiotics in patients after surgeries [7,8,10]. However, they do not focus on short-term observations regarding the quality of life of patients after surgery and changes in fecal examination [7,9]. It is worth noting that prebiotics can be used to improve the quality of GM and, furthermore, to minimize the chances of experiencing adverse symptoms after surgery. The literature provides reports on the use of the mushroom *Hericium erinaceus* and its positive effects on gut microbiota [11,12,13]. Vigna et al. demonstrated that *H. erinaceus* supplementation could decrease depression, anxiety, and sleep disorders in patients with obesity [11].

## 2. Materials and Methods

### 2.1. Study Aims

The primary aim of this study is to evaluate the impact of probiotic supplementation on postoperative gastrointestinal symptoms in patients undergoing SG. Specifically, this study aims to assess the effect of probiotics on the frequency and ease of defecation in the early weeks following surgery and determine whether probiotic supplementation reduces the incidence of constipation after SG. The secondary aim is to evaluate laboratory and stool test results to identify any biochemical and microbiological changes associated with probiotic use. The stool examination was conducted to determine whether changes in gut microbiota reflect the symptoms reported by the patients.

### 2.2. Study Design

This is a prospective, triple-blind, placebo-controlled study involving patients undergoing sleeve gastrectomy (SG) at a single center between March and April 2024. The postoperative consulting surgeon, the laboratory staff performing stool tests, and the patient did not know whether it was the probiotic or a placebo. This study included patients over 18 years old who qualified for bariatric surgery and who had no specific symptoms of gastrointestinal tract diseases. The exclusion criteria were gastrointestinal tract diseases, the use of prokinetics and proton pump inhibitors, antibiotics supplements, such as probiotics, prebiotics, or other nutritional supplements at least 4 weeks before the study, and refusal to participate. Conditions that did not disqualify participants included type 2 diabetes, hypertension, gout, sleep apnea, and hypothyroidism. Demographic and clinical data were collected during personal visits, including age, sex, weight, height, waist circumference, presence of obesity-related diseases, medications, and gastrointestinal symptoms. The outcomes of bariatric surgery were described according to the standardized outcomes reporting, including the percentage of total weight loss (%TWL) and percentage of excess weight loss (%EWL) [14].

### 2.3. Allocation

The allocation of participants was performed in a blinded manner. Patients were divided into two groups: Probiotics and Controls. The probiotics and control packages were provided by a company (Probios Ltd., Olsztyn, Poland) in identical packages labeled with random numbers. This study was unblinded after one month, once all patients had completed the follow-up.

### 2.4. Treatment

The administered substances, both probiotic and control, were in the form of disposable sachets to be dissolved in water. Based on a literature review and our own research, it was decided to use unique strains of bacteria from the Probios Ltd. collection strains of Lactobacillus plantarum AMT14 5 × 10^8^ colony-forming unit per gram (cfu/g), Bifidobacterium animalis AMT30 1 × 10^10^ cfu/g and Bifidobacterium breve AMT32 1 × 10^10^ cfu/g. The product is a modification of the existing laBIFID^®^, called laBIFID plus^®^. They have been deposited in the Polish Collection of Microorganisms at the Institute of Immunology and Experimental Therapy of the Polish Academy of Sciences in Wrocław. The strains have Polish, European, and American patents. The placebo sachets were neutral, consisting only of starch, and matched the probiotics in amount and color. Both were recommended to be refrigerated and taken once a day with the first meal. During the observation period, all patients additionally consumed the same recommended medications: low molecular weight heparin (0.4 mL subcutaneously for 10 days), omeprazole (40 mg orally for 30 days), ursodeoxycholic acid (600 mg orally for 90 days), and the multivitamin Floradix^®^ (orally for the entire period). Patients were required to follow a diet based on the same principles, monitored by a hospital dietitian. They were also instructed to drink 1.5–2 L of fluids daily.

### 2.5. Assessment of Gastrointestinal Symptoms

To assess digestive system symptoms, patients completed specially prepared questionnaires at three time points: before surgery, one month after surgery, and three months after surgery. The questionnaire contained four questions:a.The number of stools passed per week.b.Whether constipation occurred (yes or no).c.Subjective assessment of the ease of defecation.d.Feeling of bowel movement completeness.

A Likert scale of 1–5 was used for the last two questions, where 1 indicated great difficulty with defecation and a complete feeling of incomplete defecation, and 5 indicated no difficulty with defecation and a complete feeling of defecation.

Laboratory and stool tests were collected on an empty stomach one day before surgery and one month after surgery.

### 2.6. Stool Examination

The research material included stool samples collected from patients in sterile containers and transported to the microbiological laboratory in refrigerated conditions. Stool collection was performed the day before surgery and one month after surgery. The following species of bacteria and yeasts were determined in stool samples: *Candida krusei*, *Candida albicans*, *Candida tropicalis*, *Candida glabrata*, *Candida* spp., *Clostridioides difficile*, *Clostridium perfingens*, *Clostridium* spp., Vancomycin-resistant enterococci, *Enterococcus* spp., *Enterococcus faecium*, *Enterococcus faecalis*, *Escherichia coli*, Carbapenemase-producing *Escherichia coli* (CPE), *Klebsiella pneumoniae*, Carbapenemase-producing *Klebsiella pneumoniae* (CPK), *Pseudomonas aeruginosa*, *Staphylococcus aureus*, and *Staphylococcus saprophyticus*, and the total number of bacteria grown in anaerobic conditions and aerobic conditions. Detailed characteristics of the media for growing given bacterial species are included in Appendix A.

After preparation of the stock suspension and decimal dilutions, stool samples were taken and inoculated on media. The samples were incubated under temperature conditions and time according to the substrate manufacturer’s recommendations. Immediately after incubation, colonies characteristic of a given genus/species were counted using a stereoscopic microscope (SZX–ILLK 200, Olympus Optical, Tokyo, Japan), after which the morphology of all cells was checked microscopically (Nikon Microphot-FXA, Tokyo, Japan) to identify colonies of quantifiable dilution and confirm genus and species affiliation.

### 2.7. Surgical Technique

The surgical techniques were performed in accordance with established guidelines [15]. SG was performed using a bougie size of 36F, starting 4–6 cm from the pylorus.

### 2.8. Statistical Analysis

A descriptive statistical analysis was conducted. All data were analyzed using Statistica software 13.PL (StatSoft Inc., Tulsa, OK, USA). The normal distribution was checked using the Shapiro–Wilk test. Numbers and percentages were used for categorical variables. For continuous variables with a normal distribution, the mean and standard deviation were used. Student’s t-test was applied for independent variables. In cases where continuous variables showed a non-normal distribution, the Mann–Whitney U test was utilized to compare the two groups. For categorical variables, we used the chi-square test. A *p*-value of <0.05 was considered statistically significant.

## 3. Results

A total of 31 patients were analyzed in this study. A total of 15 patients were in the Probiotics group, and 16 patients were in the Controls group. The patients between groups did not differ in characteristics, Table 1. The mean ages were 40.9 years and 40.9 years, and the mean BMIs were 43.2 kg/m^2^ and 40.1 kg/m^2^, respectively. There were no adverse effects associated with the use of probiotics. There were no complications in this group of patients.

### 3.1. Postoperative Gastrointestinal Symptoms

Patients’ subjective feelings regarding the functioning of their digestive system differed statistically significantly between the Probiotics and Controls groups: Table 2. Patients in the Probiotics group passed significantly more stools per week than patients in the Controls group (*p* = 0.027). Only one patient (0.6%) supplementing Probiotics reported the occurrence of constipation, while nine patients (56.3%) in the Controls group had constipation at the time of follow-up (*p* = 0.002). Patients in the Probiotics group presented a significantly easier feeling of defecation and greater bowel movement completeness compared to Patients in the Controls group (*p* = 0.015, *p* = 0.004, respectively).

### 3.2. Outcomes of Bariatric Surgery

One month after the surgery, patients in the Probiotics group and the Controls group did not differ in weight loss, Table 1. The %TWL were 8.6% and 8.9%, respectively (*p* = 0.772). The occurrence of constipation did not affect the change in waist circumference (*p* = 0.465). There were no statistically significant changes in laboratory tests after the surgery among patients in both groups, Table 3.

### 3.3. Stool Microbiota

A total of 14 patients in the Probiotics group and 14 patients in the Controls group provided stool samples one month after the surgery. The follow-up rate was 90.3%. Numerically, the content of bacteria in stool was not statistically different in patients in both study groups: Appendix A. Taking into account common laboratory standards, there were higher increases in *Enterococcus faecalis* and *Clostridium perfringers* in the stools of patients in the Controls group compared to the Probiotics group. In the Probiotics group, some patients’ levels of *Enterococcus faecium* dropped to normal levels after being elevated before surgery.

## 4. Discussion

Our study aimed to evaluate the impact of probiotics on patients undergoing SG, focusing on gastrointestinal symptoms, weight loss, and laboratory outcomes. The results indicate that, while probiotics do not significantly affect weight loss or laboratory parameters, they do substantially reduce gastrointestinal discomfort in the early postoperative weeks. This finding is particularly relevant given that postoperative gastrointestinal symptoms can significantly affect patient recovery and overall quality of life. There are studies that examine the effects of probiotics on bariatric surgery outcomes. However, most of them involve bypass procedures, mainly RYGB. Therefore, our paper, although on a small group, seems to be important because it also shows the results of probiotics after the most frequently performed bariatric surgery in the world, which is SG.

Our finding that probiotics significantly reduce gastrointestinal discomfort aligns with multiple studies emphasizing the beneficial effects of probiotics on gastrointestinal function [16,17,18,19]. As previously mentioned, most studies concern the RYGB, where the change in bacterial flora and the resulting gastroenterological problems are caused by a change in the anatomy of the digestive tract [17,18,19,20]. Swierz et al. noted short-term improvements in gastrointestinal symptoms without a meaningful impact on weight loss or quality of life [16]. These improvements are crucial for patient recovery and comfort, as they can lead to a smoother postoperative experience and potentially reduce hospital readmission rates. Sherf-Dagan et al. noted that while probiotics did not significantly alter hepatic or inflammatory outcomes, the reduction in gastrointestinal symptoms could enhance overall patient well-being during the critical postoperative period [21]. Moreover, the reduction in gastrointestinal symptoms, such as constipation, aligns with our findings and underscores the potential of probiotics in managing postoperative discomfort effectively.

Our study did not find significant differences in weight loss between the Probiotics and Controls groups, which is consistent with previous research [16,19]. The meta-analysis by Wang et al. suggested that probiotics might improve lipid metabolism and reduce weight and food intake [19]. These effects were not universally observed across all studies, indicating that the impact of probiotics on weight loss may depend on various factors such as the type of probiotic strains used, duration of supplementation, and patient characteristics [19]. Ramos et al. presented a positive effect on the plasma metabolite profile after RYGB [22]. Kazzi et al. demonstrated a reduction in gastrointestinal symptoms and improvements in biochemical markers such as triglycerides and liver enzymes among patients receiving probiotics after SG [17]. These findings suggest that, while probiotics may not significantly influence weight loss, they could play a role in improving gastrointestinal health and modulating key biochemical markers, which could have long-term benefits for metabolic health.

The role of probiotics in modulating gut microbiota post-bariatric surgery is becoming the subject of more and more extensive research [23,24]. Cabanillas-Lazo et al. highlighted the significant changes in gut microbiota composition following bariatric procedures, suggesting potential benefits from targeted probiotic interventions [23]. Komorniak et al. illustrated specific microbial changes, noting the presence of beneficial bacteria such as *Christensenellaceae* and *Oscillibacter* in SG patients, which might contribute to improved gastrointestinal health [25]. These findings support the notion that probiotics can positively influence gut microbiota, thereby enhancing gastrointestinal function and reducing symptoms such as constipation.

Other studies have suggested that the effect of probiotics on weight loss may depend on factors such as the type of probiotic species used, the strain, and the duration of consumption, as well as the type of bariatric surgery performed. Woodard et al. demonstrated short-term benefits of *Lactobacillus* supplementation after RYGB, but these effects were only statistically significant in the early postoperative period [26]. Similarly, Mokhtari et al. reported benefits with probiotic supplementation during the first three months post-surgery, but these benefits diminished without further supplementation [27]. This variability in outcomes emphasizes the need for more research to determine the optimal type and duration of probiotic use in different surgical contexts.

There are studies in the literature that attribute the occurrence of constipation to certain strains of bacteria [28,29]. In our study, patients who did not receive probiotics and complained of constipation had an increase in bacteria of *E. faecalis*, *E. facium*, and *C. perfringers*. In patients receiving probiotics, a return to normal in these bacteria was observed. There are no clear data on the cause of flatulence and constipation caused by an excess of these bacteria, and further research on a larger group of patients is required.

While some systematic reviews report modest effects of probiotics on weight loss, the overall evidence remains inconsistent due to factors such as study heterogeneity, probiotic strain differences, and methodological limitations [16,19,24]. Furthermore, the effect of diet and other confounding factors, such as medication use (antibiotics and proton pump inhibitors), remains underexplored in many studies. Our findings add to this body of literature by highlighting the challenges in assessing the long-term effects of probiotics after SG and the need for standardized protocols in future research.

A limitation of our study is the relatively small number of patients, which limits the generalizability of the findings. Larger studies are needed to confirm our results and provide more robust evidence regarding the effects of probiotics on this patient population. Moreover, the short follow-up period may not capture the long-term effects of probiotic supplementation on weight loss and metabolic outcomes. While we observed significant improvements in gastrointestinal symptoms, this study did not thoroughly investigate other potential benefits or adverse effects of probiotics, such as their impact on immune function, psychological well-being, or long-term metabolic health. Future research should aim to address these gaps to provide a more comprehensive understanding of the role of probiotics in postoperative care.

## 5. Conclusions

The clinical implications of our findings suggest that, while probiotics may not significantly impact weight loss or laboratory outcomes, their role in reducing postoperative gastrointestinal symptoms warrants consideration. Given the lack of adverse effects reported in our study and others, incorporating probiotics into postoperative care protocols could enhance patient comfort and recovery. Therefore, using the analyzed formula may equip specialists with a tool to stabilize patients after SG.

## Figures and Tables

**Table 1 nutrients-16-03498-t001:** Characteristics of patients before and one month after the surgery. (BMI body mass index, %TWL percentage of total weight loss, %EWL percentage of excess weight loss).

Value	Probiotics	Controls	*p*
Age [years]	40.9 ± 10.6	40.9 ± 11.9	0.998
BMI before surgery [kg/m^2^]	43.2 ± 4.7	40.1 ± 3.8	0.062
BMI after surgery [kg/m^2^]	39.4 ± 4.3	36.5 ± 3.6	0.056
%TWL [%]	8.6 ± 2.1	8.9 ± 1.9	0.722
%EWL [%]	21.4 ± 5.9	24.5 ± 6.4	0.186
Change in waist circumference [cm]	8.9 ± 2.8	7.8 ± 4.6	0.465

**Table 2 nutrients-16-03498-t002:** Answers to the questionnaire regarding passing stools after surgery.

	Probiotics	Controls	*p*
Median number of stools passed per week	7	3	0.027 *
Occurrence of constipation	0.6% (1 patient)	56.3% (9 patients)	0.002 **
Median feeling of ease of defecation	5	4	0.015 *
Median feeling of bowel movement completeness	5	4	0.004 *

* Mann–Whitney U test; ** chi-square test.

**Table 3 nutrients-16-03498-t003:** Laboratory tests after surgery. (* refers to values after surgery).

	Baseline		After Surgery		*p **
Value	Probiotics	Controls	Probiotics	Controls	
Hemoglobin [g/dL]	14.3 ±1.2	13.8 ±0.7	13.9 ± 1.0	13.6 ± 1.0	0.333
Ferritin [mg/dL]	139 ± 131	90.3 ± 84.4	153 ± 127	125 ± 88.7	0.496
Albumin [kg/m^2^]	45.1 ± 1.7	44.7 ± 2.5	45.0 ± 1.9	43.8 ± 2.7	0.171
Protein [%]	7.72 ± 0.5	7.6 ± 0.5	7.23 ± 0.4	7.1 ± 0.4	0.359
Glycated hemoglobin [%]	5.7 ± 0.3	6.0 ± 0.8	5.4 ± 0.3	5.7 ± 0.5	0.142
Total cholesterol [mg/dL]	200 ± 37	199 ± 28	178 ± 33	172 ± 37	0.655
Triglicerydes [mg/dL]	144 ± 33	118 ± 54	144 ± 66	111 ± 30.6	0.100
Vitamin B12 [pg/mL]	458 ± 146	510 ± 211	508 ± 112	650 ± 402	0.213

## Data Availability

The original contributions presented in this study are included in the article/Appendix A, further inquiries can be directed to the corresponding authors.

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
