# Peer review of "Initial Study on the Impact of Probiotics on Postoperative Gastrointestinal Symptoms and Gut Microbiota after Sleeve Gastrectomy: A Placebo-Controlled Study"

_nutrients, 2024, doi:10.3390/nu16203498_

Round 1
Reviewer 1 Report
Comments and Suggestions for Authors
This "RCT" aimed to examine the impact of a probiotic on gastrointestinal symptoms following sleeve gastrectomy.
The formal presentation of the manuscript needs a complete revision. The authors must strictly adhere to the CONSORT guidelines, which should be applied to the manuscript and attached as supplementary material.
In my opinion, this work should be converted into a brief report, as it does not seem to have the quality to be defined as an RCT nor original research. It should be presented as a proof-of-concept or pilot study, removing any reference to the word "trial". The statistical analyses need to be completely revised, as they are incorrect in terms of the tests chosen and their formal presentation.
Major comments:
1) The concept of RCT conflicts with "initial study" in the title. An RCT should have adequate statistical power calculated beforehand to ensure the credibility and sustainability of the results. A pilot study claiming to be an RCT becomes difficult to justify;
2) Malabsorption of micro- and macronutrients is not a "symptom" to be listed, but rather a complication that causes signs and, potentially, symptoms. This sentence needs to be corrected;
3) The introduction is missing some data. I would suggest adding that, besides probiotics, prebiotics (to expand the scope of nutraceutical interventions in this context) such as mushrooms (e.g., Hericium erinaceus) have shown gastrointestinal benefits (https://pubmed.ncbi.nlm.nih.gov/37465689/) as well as effects on obesity (https://pubmed.ncbi.nlm.nih.gov/31084539/, https://pubmed.ncbi.nlm.nih.gov/31118969/). I recommend citing these three studies for completeness;
4) I would remove the aims of the study from the introduction. Given the study design, these deserve a dedicated paragraph in section 2 of the materials and methods;
5) As this is an RCT, the inclusion and exclusion criteria seem too few and poorly specified. These criteria should ensure the absence of bias. How was the exclusion of gastrointestinal disorders carried out? Did you exclude only those taking antibiotics? What about those who had recently taken probiotics, prebiotics, postbiotics, symbiotics, or nutraceuticals? What about those on prokinetics, antacids, or antispasmodics? Were patients with other significant comorbidities that could cause gastrointestinal symptoms excluded (e.g., those with thyroid disorders or adrenal diseases)? The criteria need to be completely removed and rewritten from scratch. In their current form, they give rise to numerous biases;
6) The collected data deserve a separate paragraph, and it should be explained how they were calculated. Again, this is a trial, and everything must be written in detail;
7) Lines 71,72: "provided by a company" means nothing, and this must be clarified;
8) In paragraph 2.4, it is unclear what scale the authors used. How was constipation defined? How was the "subjective assessment of the ease of defecation" defined, and how was the "feeling of bowel movement completeness" defined? Was Cronbach's alpha tested, considering this appears to be an unvalidated, non-standardized questionnaire invented by the authors?
9) Section 2.6: once again, a lack of details. Which guidelines are being referred to? They need to be cited;
10) Can an RCT have a statistical analysis section of just 6 lines? There is no mention of a power calculation or sample size estimation. Again, the authors claim the study is an RCT;
11) How can a trial with 31 patients present variables as means and SD? The data distribution will almost certainly be non-normal with this sample size. Median and IQR should have been considered;
12) How can a Student's t-test be used on 31 patients? It is a parametric test designed for numerical variables with a normal distribution. The test is completely inappropriate;
13) Is the p-value one-tailed or two-tailed?
14) Table 2 is unreadable. What are the numbers? Means? How was the onset of constipation assessed, as it is a categorical variable? What test was used? A footnote should be added to the table to clarify all these aspects. It's also unclear why the authors present the table this way when they used a Likert scale;
15) I am surprised, based on Table 3, that no laboratory parameters changed after sleeve gastrectomy, as there are no significant differences in any case;
16) Table 4 should be completely removed. The microbiota analysis should be presented in a SERIOUS manner, with graphs, plots, and detailed tables;
17) The discussion is overly simplistic.
Minor comments:
1) In line 36, there is a misplaced parenthesis for the acronym "SG";
2) Bacterial names should be italicized;
3) There is an extra full stop in line 259.

Author Response
This "RCT" aimed to examine the impact of a probiotic on gastrointestinal symptoms following sleeve gastrectomy.
The formal presentation of the manuscript needs a complete revision. The authors must strictly adhere to the CONSORT guidelines, which should be applied to the manuscript and attached as supplementary material.
In my opinion, this work should be converted into a brief report, as it does not seem to have the quality to be defined as an RCT nor original research. It should be presented as a proof-of-concept or pilot study, removing any reference to the word "trial". The statistical analyses need to be completely revised, as they are incorrect in terms of the tests chosen and their formal presentation.
We thank the reviewer for thoroughly analyzing our work. We agree that, as an RCT, our patient group is small, and this represents a significant limitation. We will present our initial results as a placebo-controlled study. We hope that all the changes we have made will be understood and will meet the reviewer’s expectations. We believe that these revisions will make the manuscript clearer and improve its quality.
Major comments:
1) The concept of RCT conflicts with "initial study" in the title. An RCT should have adequate statistical power calculated beforehand to ensure the credibility and sustainability of the results. A pilot study claiming to be an RCT becomes difficult to justify;
Thank you for this comment. We agree with the reviewer, and we have removed all references to the RCT nature of the study from our article, leaving it as an initial placebo-controlled study.
2) Malabsorption of micro- and macronutrients is not a "symptom" to be listed, but rather a complication that causes signs and, potentially, symptoms. This sentence needs to be corrected;
Thank you for this comment. We have revised the sentence to avoid any confusion (line 39).
3) The introduction is missing some data. I would suggest adding that, besides probiotics, prebiotics (to expand the scope of nutraceutical interventions in this context) such as mushrooms (e.g., Hericium erinaceus) have shown gastrointestinal benefits (https://pubmed.ncbi.nlm.nih.gov/37465689/) as well as effects on obesity (https://pubmed.ncbi.nlm.nih.gov/31084539/, https://pubmed.ncbi.nlm.nih.gov/31118969/). I recommend citing these three studies for completeness;
Thank you for this comment. The reviewer rightly pointed out that prebiotics represent another aspect of research that should be conducted, especially in the context of patient prehabilitation and preparation for surgery. We have added a section regarding prebiotics to the introduction (lines: 50-55).
4) I would remove the aims of the study from the introduction. Given the study design, these deserve a dedicated paragraph in section 2 of the materials and methods;
Thank you for this comment. We have developed a dedicated paragraph for study aims (2.1).
5) As this is an RCT, the inclusion and exclusion criteria seem too few and poorly specified. These criteria should ensure the absence of bias. How was the exclusion of gastrointestinal disorders carried out? Did you exclude only those taking antibiotics? What about those who had recently taken probiotics, prebiotics, postbiotics, symbiotics, or nutraceuticals? What about those on prokinetics, antacids, or antispasmodics? Were patients with other significant comorbidities that could cause gastrointestinal symptoms excluded (e.g., those with thyroid disorders or adrenal diseases)? The criteria need to be completely removed and rewritten from scratch. In their current form, they give rise to numerous biases;
Thank you for this comment. We have expanded the exclusion criteria to ensure clarity (2.2). In the study, we excluded patients with any gastrointestinal disorders, including those requiring medication, probiotics/prebiotics, or prokinetics, as well as PPIs. A four-week washout period was applied for antibiotics or other medications. Additionally, each patient was prepared for surgery with the same diet, supervised by a hospital dietitian. Conditions that did not disqualify participants included diabetes, hypertension, gout, sleep apnea, and hypothyroidism.
6) The collected data deserve a separate paragraph, and it should be explained how they were calculated. Again, this is a trial, and everything must be written in detail;
Thank you for that comment. We provided a more detailed description of the data obtained during the visits. Additionally, in a separate paragraph 2.5, we thoroughly explained how the data regarding subjective symptoms were collected.
7) Lines 71,72: "provided by a company" means nothing, and this must be clarified;
Thank you for that comment. We have provided more data to clarified it.
8) In paragraph 2.4, it is unclear what scale the authors used. How was constipation defined? How was the "subjective assessment of the ease of defecation" defined, and how was the "feeling of bowel movement completeness" defined? Was Cronbach's alpha tested, considering this appears to be an unvalidated, non-standardized questionnaire invented by the authors?
Thank you for this comment. We agree with the reviewer that the three questions regarding bowel movements are subjective and lack standardization. Therefore, our study should have been presented as an initial study, rather than as an RCT, as it was. Constipation was assessed by patients in a binary (yes/no) format, while the two questions concerning ease of defecation and assessment of bowel movement completeness were rated using a 1-5 Likert scale. The study included patients who did not report constipation before surgery and who rated their bowel movements as 5 on the 1-5 scale.
9) Section 2.6: once again, a lack of details. Which guidelines are being referred to? They need to be cited;
Thank you for that comment. We have added the reference.
10) Can an RCT have a statistical analysis section of just 6 lines? There is no mention of a power calculation or sample size estimation. Again, the authors claim the study is an RCT;
Thank you for that comment. We agree with the reviewer that calling the study an RCT was not correct, and we have revised the format of our work accordingly.
11) How can a trial with 31 patients present variables as means and SD? The data distribution will almost certainly be non-normal with this sample size. Median and IQR should have been considered; 12) How can a Student's t-test be used on 31 patients? It is a parametric test designed for numerical variables with a normal distribution. The test is completely inappropriate;
Thank you for this comment. Of course, before starting the calculations, we performed the Shapiro-Wilk test for all variables (the results are below in the table). The distribution is normal, so we decided to proceed with the t-test.
Variable |
Probiotics |
Controls |
Age |
0.689 |
0.115 |
BMI before surgery |
0.681 |
0.137 |
BMI after surgery |
0.305 |
0.146 |
%TWL |
0.859 |
0.277 |
%EWL |
0.591 |
0.987 |
Change in waist circumference |
0.071 |
0.472 |
13) Is the p-value one-tailed or two-tailed?
Thank you for that comment. We used two-tailed test.
14) Table 2 is unreadable. What are the numbers? Means? How was the onset of constipation assessed, as it is a categorical variable? What test was used? A footnote should be added to the table to clarify all these aspects. It's also unclear why the authors present the table this way when they used a Likert scale;
Thank you for that comment. Of course, these parameters were analyzed using a different test than the t-test, which was definitely lacking in the methodology. In cases where continuous variables showed a non-normal distribution, the Mann-Whitney U test was utilized to compare the two groups. For categorical variables, we used the chi-square test. We added a section to the statistical analysis as well as a footnote indicating which tests were performed.
15) I am surprised, based on Table 3, that no laboratory parameters changed after sleeve gastrectomy, as there are no significant differences in any case;
Thank you for this comment. In the first month after surgery, we rarely observe changes in these parameters in patients who did not have any pre-existing disorders. The first deficiencies typically appear around six months post-operation, if they do occur. However, we have different observations with bypass procedures such as RYGB, OAGB, and SASI, where these changes are noticeable in an earlier period.
16) Table 4 should be completely removed. The microbiota analysis should be presented in a SERIOUS manner, with graphs, plots, and detailed tables;
Thank you for that comment. We believe that the way our results are presented in Table 4 is incorrect and should not be included in the paper. Full stool and individual microbiota tests have been attached in Supplement 2.
17) The discussion is overly simplistic.
Thank you for that comment. We have expanded the discussion with several elements that we hope provide a broader perspective on the topic. The limitations of studies on probiotics include significant heterogeneity, the use of different bacterial strains, varying patient groups, and different inclusion and exclusion criteria. Therefore, we believe that in future research, it is important to establish a group that would conduct studies on the standardization of incorporating probiotics into perioperative care.
Minor comments:
1) In line 36, there is a misplaced parenthesis for the acronym "SG";
Thank you for that comment. We have corrected that.
2) Bacterial names should be italicized;
Thank you for that comment. We have corrected that.
3) There is an extra full stop in line 259.
Thank you for that comment. We have corrected that.
Reviewer 2 Report
Comments and Suggestions for Authors
The manuscript describes initial study on the impact of probiotic on postoperative gastro- intestinal symptoms after sleeve gastrectomy: a randomized controlled trial. The topic is relevant to the aim and scope of the Nutrients. The manuscript is well written and easy to follow. However, this manuscript meets the standard for acceptance after addressing the below comment:
The title does not match with stool examination. Please explain in the manuscript why the stool examination is necessary for the study on the impact of probiotic on postoperative gastrointestinal symptoms after sleeve gastrectomy: a randomized controlled trial.

Author Response
The manuscript describes initial study on the impact of probiotic on postoperative gastro- intestinal symptoms after sleeve gastrectomy: a randomized controlled trial. The topic is relevant to the aim and scope of the Nutrients. The manuscript is well written and easy to follow. However, this manuscript meets the standard for acceptance after addressing the below comment:
We thank the reviewer for thoroughly analyzing our work. We appreciate the recognition of our work and the positive feedback. We hope that the changes we have made will lead to the acceptance of the paper.
The title does not match with stool examination. Please explain in the manuscript why the stool examination is necessary for the study on the impact of probiotic on postoperative gastrointestinal symptoms after sleeve gastrectomy: a randomized controlled trial.
Thank you for that comment. The stool examination was conducted to determine whether changes in gut microbiota reflect the symptoms reported by the patients. Although we did not obtain statistically significant results, which may be due to the small sample size, the study shows a certain trend. There were higher increases in Enterococcus faecalis and Clostridium perfringens in the stools of patients in the Controls group compared to the Probiotics group. In the Probiotics group, some patients’ levels of Enterococcus faecium dropped to normal levels after being elevated before surgery. However, larger studies are needed to prove the association of these bacteria with symptoms. We have modified the title and added the explanation in aims of the study.
Reviewer 3 Report
Comments and Suggestions for Authors
This manuscript is written well.
I believe this study is well-written, particularly in terms of its design, subject selection, and discussion of the results. I hope this study will serve as a foundation for further research, while also considering its limitations.

This manuscript will be able to require for minor editing of English language scurvies.
Author Response
This manuscript is written well.
I believe this study is well-written, particularly in terms of its design, subject selection, and discussion of the results. I hope this study will serve as a foundation for further research, while also considering its limitations.
We thank the reviewer for analyzing our work. We appreciate the positive feedback and recognition of our efforts.
Round 2
Reviewer 1 Report
Comments and Suggestions for Authors
The authors have significantly revised the work and downplayed its trial nature. Under these conditions, the work has improved.
Reviewer 2 Report
Comments and Suggestions for Authors
The manuscript is sufficient for publication.